# Differential Effects of Histone Deacetylases on the Expression of NKG2D Ligands and NK Cell-Mediated Anticancer Immunity in Lung Cancer Cells

**DOI:** 10.3390/molecules26133952

**Published:** 2021-06-28

**Authors:** Haeryung Cho, Woo-Chang Son, Young-Shin Lee, Eun Jung Youn, Chi-Dug Kang, You-Soo Park, Jaeho Bae

**Affiliations:** 1Department of Biochemistry, Pusan National University School of Medicine, Yangsan 50612, Korea; chohr9665@naver.com (H.C.); dudtls0701@naver.com (Y.-S.L.); youndmswjd@naver.com (E.J.Y.); kcdshbw@pusan.ac.kr (C.-D.K.); 2PNU GRAND Convergence Medical Science Education Research Center, Pusan National University School of Medicine, Yangsan 50612, Korea; 3Department of Research Center, Dongnam Institute of Radiological and Medical Sciences, Gijang, Busan 46033, Korea; 75dokdo@naver.com

**Keywords:** HDAC, lung cancer, NK cell, NKG2D ligands

## Abstract

Histone acetylation is an epigenetic mechanism that regulates the expression of various genes, such as natural killer group 2, member D (NKG2D) ligands. These NKG2D ligands are the key molecules that activate immune cells expressing the NKG2D receptor. It has been observed that cancer cells overexpress histone deacetylases (HDACs) and show reduced acetylation of nuclear histones. Furthermore, HDAC inhibitors are known to upregulate the expression of NKG2D ligands. Humans have 18 known HDAC enzymes that are divided into four classes. At present, it is not clear which types of HDAC are involved in the expression of NKG2D ligands. We hypothesized that specific types of HDAC genes might be responsible for altering the expression of NKG2D ligands. In this study, we monitored the expression of NKG2D ligands and major histocompatibility complex (MHC) class I molecules in lung cancer cells which were treated with six selective HDAC inhibitors and specific small interfering RNAs (siRNAs). We observed that treatment with FK228, which is a selective HDAC1/2 inhibitor, also known as Romidepsin, induced NKG2D ligand expression at the transcriptional and proteomic levels in two different lung cancer cell lines. It also caused an increase in the susceptibility of NCI-H23 cells to NK cells. Silencing HDAC1 or HDAC2 using specific siRNAs increased NKG2D ligand expression. In conclusion, it appears that HDAC1 and HDAC2 might be the key molecules regulating the expression of NKG2D ligands. These results imply that specifically inhibiting HDAC1 and HDAC2 could induce the expression of NKG2D ligands and improve the NK cell-mediated anti-cancer immunity.

## 1. Introduction

Lung cancer is one of the most common cancers, and has the highest mortality among various types of cancer worldwide [1]. Despite the improvements in cancer treatments over the last few decades, lung cancer has a poor prognosis due to a high relapse rate and distant metastasis. It is known that host immunity against cancer is one of the most important factors determining the survival of patients with lung cancer. Consequently, it could be suggested that enhancing the activity of immune cells might improve the prognosis of patients with lung cancer [2]. Moreover, it is known that NK cells are involved in the immune surveillance of cancer and that persistent NK cell activity is associated with a good prognosis for patients with lung cancer [3]. Unlike T cells, a balance between the signals from inhibitory and activating receptors regulates the activity of NK cells, independently of exposure to antigens. Natural killer group 2, member D (NKG2D) is one of the major activating receptors that is expressed by human NK cells and a few types of T cell; it transduces activating signals to immune cells upon binding to the NKG2D ligands on cancer cells [4]. Moreover, NKG2D ligands are rarely expressed by normal cells, and their expression is increased by infected and transformed cells [5,6]. However, it was reported that the expression of NKG2D ligands in established cancer cells might not be sufficient to induce NK cell-mediated immune responses [7,8]. Additionally, the immunosuppressive microenvironment in patients is thought to frequently suppress NK cell activation [9]. A better understanding of the mechanisms underlying the regulation of NKG2D ligands might have implications for the development of therapeutic strategies that can increase the expression of NKG2D ligands to levels that are adequate for NK cell activation. In humans, NKG2D ligands are encoded by eight genes, which are regulated at both the transcriptional and post-transcriptional levels [10]. Histone deacetylation is an epigenetic process that affects gene transcription, and was reported to be one of the key mechanisms regulating the transcription of genes encoding the NKG2D ligands [11].

HDACs are encoded by 18 different genes, which are divided into several classes based on sequence homology and cofactor requirements. Table 1 shows the classification of HDACs. Although previous studies have reported a significant association between the acetylation levels of nuclear histones and the transcription of NKG2D ligands [12], it is still not known which types of histone deacetylase are involved in the regulation of NKG2D ligands. Therefore, we aimed to ascertain which types of HDAC were responsible for regulating the expression of NKG2D ligands using selective HDAC inhibitors. Finally, we examined whether the selective inhibitors could enhance the cytotoxic effects of NK cells against lung cancer cells.

## 2. Results

### 2.1. The HDAC1/2 Inhibitor, FK228, Increased Transcription of NKG2D Ligands in Lung Cancer Cells

The transcription of NKG2D ligands, namely, MHC class I polypeptide-related chain proteins A (MICA), MICB, UL16 binding protein 1 (ULBP1), ULBP2, and ULBP3, was analyzed after treating lung cancer cells with five types of HDAC inhibitor for 18 h. It was observed that FK228, RGFP966, and PCI34051 selectively inhibited the class I HDACs HDAC1/2, HDAC3, and HDAC8, respectively [13,14,15]. The selective inhibitor MC1568 targets HDAC4/5, which are class IIa HDACs, whereas Tubacin is an inhibitor of HDAC6, which belongs to HDAC class IIb [16,17]. Treating NCI-H23 and A549 lung cancer cells with FK228 led to an increase in the transcription of the five NKG2D ligands: MICA, MICB, ULBP1, ULBP2, and ULBP3. The increase in MICA and ULBP1 expression in NCI-H23 and A549 cells, respectively, following FK228 treatment, was dose-dependent (Figure 1A,F). Specific concentrations of other HDAC inhibitors also affected the transcription of NKG2D ligands in NCI-H23 cells (Figure 1B–E). Moreover, moderate concentrations of RGFP966 (10 or 100 nM) led to a decrease in the transcription of MICB, ULBP1, and ULBP3. The MC1568 decreased ULBP2 transcription at a concentration of 1 nM. Treating cells with 10 and 1000 nM of Tubacin decreased the transcription of ULBP2 and ULBP1, respectively. Finally, we observed that 1, 10, and 1000 nM of PCI34051 decreased the expression of ULBP3, ULBP2, and MICB, respectively. These results suggest that amongst the inhibitors that were tested, only FK228, the HDAC1/2 inhibitor, could induce the expression of NKG2D ligands.

### 2.2. Surface Expression of NKG2D Ligands in Lung Cancer Cells Is Increased by FK228 Treatment

Lung cancer cells were treated with five types of HDAC inhibitor, following which, the surface expression of NKG2D ligands and MHC class I molecules was analyzed via flow cytometry. Amongst the five NKG2D ligands, the surface expression of ULBP1 and ULBP2 increased by 1.9- and 10.2-fold, respectively, in NCI-H23 cells that were treated with 100 nM FK228 (Figure 2A). Treating A549 cells with FK228 increased the expression of MICA, MICB, ULBP1, ULBP2, and ULBP3, by 5.6-, 2.3-, 2.9-, 11.6-, and 1.4-fold, respectively. The expression of MHC class I inhibitory ligands also increased by 1.3-fold in FK228-treated A549 cells. We observed that treatment with Tubacin and PCI34051 caused a significant reduction in the surface expression of the ULBP1 and ULBP2 ligands, despite the discrepancies between the levels of transcripts and protein expression on the cell surface (Figure 2B–E). These discrepancies were attributed to unknown post-transcriptional mechanisms. The absence of ULBP3 in NCI-H23 cells was marked as ND in the figures.

### 2.3. Silencing HDAC1 or HDAC2 Using Small Interfering RNA (siRNA)

The siRNAs that were transfected into NCI-H23 and A549 cells were able to effectively silence HDAC1 and HDAC2 (Figure 3A,B). To further investigate the results from our previous experiments, which showed that treatment with the HDAC1/2 inhibitor FK228 affected the expression of NKG2D ligands, we designed two siRNAs that could specifically knock down HDAC1 or HDAC2. Cells in which HDAC1 or HDAC2 were knocked down were used to confirm that each HDAC could individually alter the expression of NKG2D ligands.

### 2.4. Silencing HDAC1 and HDAC2 Increased the Surface Expression of NKG2D Ligands in Lung Cancer Cells

The surface expression of the MICB, ULBP1, and ULBP2 proteins increased in NCI-H23 cells in which HDAC1 was knocked down (Figure 4A). There was an increase in the expression of ULBP2 in cells with HDAC2 knocked down (Figure 4B). Silencing HDAC1 in A549 cells led to an increase in the surface expression of MICA, MICB, ULBP1, ULBP2, and ULBP3 proteins (Figure 4C). There was an increase in the levels of MICB, ULBP1, and ULBP3 proteins in cells where HDAC2 had been knocked down (Figure 4D). In summary, the data suggest that deleting HDAC1/2 could be an important mechanism for regulating the expression of NKG2D ligands, and that the FK228-dependent increase in expression of NKG2D ligands was mediated via the inhibition of HDAC1/2 activity.

### 2.5. The HDAC1/2 Inhibitor, FK228, Increases the Susceptibility of NCI-H23 Cells to NK Cells

To investigate whether treatment with the HDAC1/2 inhibitor, FK228, led to an increase in the NK cell-mediated lysis of cancer cells, cytotoxicity assays were performed using NCI-H23 cells. The cells were treated with 100 nM of FK228 and co-cultured with NK-92 cells. Thereafter, the proportion of dead cells was determined by propidium iodide (PI) staining (Figure 5). There was a significant increase in the susceptibility of NCI-H23 cells to NK cell-mediated lysis after treatment with FK228. This suggested that the increased susceptibility was due to the augmented expression of NKG2D ligands.

## 3. Discussion

Previous studies suggest that HDAC inhibitors could be used to stop the growth of cancer cells, because HDACs are frequently upregulated and involved in the growth of various types of cancer cell [18,19]. Histones are used to fold the DNA, and are the primary targets of HDACs and histone acetyl transferases (HATs), the two enzymes that regulate the acetylation levels of these histones. Alterations in histone acetylation can change the chromatin structure and affect the expression of various genes at the transcription level. Furthermore, the deacetylation of other proteins altered a number of biological processes, such as inflammation, cell proliferation, apoptosis, and carcinogenesis [20]. For example, a non-histone target of HDACs is p53, which is a tumor suppressor protein. Acetylated p53 induces apoptosis and autophagy in chronic myeloid leukemia cells [21,22]. The other non-histone targets of acetylation include tubulin, heat-shock proteins, NF-κB subunit p65, E2Fs, and myoblast determination proteins. The modification of these intracellular proteins is associated with the biological effects mediated by HDACs.

Based on sequence similarity and cofactor dependency, HDACs have been grouped into four classes and two families, namely, the classical zinc-dependent family and the NAD-dependent silent information regulator 2 (Sir2)-related protein (sirtuin) family. In humans, members of the classical family, i.e., HDAC1, -2, -3, and -8, are grouped as class I HDACs; HDAC4, -5, -6, -7, -9, and -10 are grouped as class II HDACs; and HDAC11 is a class IV HDAC. The seven isoforms of the sirtuin family are categorized as class III (Table 1) [23].

Thus far, many types of HDAC inhibitor have been developed, and some of them have been approved for the treatment of specific diseases. The inhibitor FK228, also known as Romidepsin, was approved by the U.S. Food and Drug Administration for the treatment of cutaneous T-cell lymphoma and peripheral T-cell lymphoma. FK228 acts as a prodrug that breaks disulfide bonds and produces free thiols. The released thiols bind to a zinc atom in the binding pockets of the zinc-dependent histone deacetylases, and selectively block the activity of HDAC1 and HDAC2 [24,25,26]. RGFP966, *N*-(o-aminophenyl) carboxamide, is a tight-binding competitive inhibitor which targets class I HDACs. It is most effective against HDAC3 [15]. RGFP966 reportedly decreases cell growth in cutaneous T-cell lymphoma cell lines by increasing apoptosis and DNA damage, and impairing progression into the S phase [27]. MC1568 facilitates proteasomal degradation of HDAC4 and HDAC5. It has been shown to arrest myogenesis through the inhibition of myocyte enhancer factor 2 and RARγ, which mediates the differentiation of muscle cells [28]. MC1568 decreases the levels of interleukin-8 and the proliferation of melanoma cells, and has shown potent and specific inhibition of class II HDACs, such as HDAC4 and HDAC5 [16]. Tubacin is known to inhibit HDAC6, which alters the levels of tubulin acetylation. It represses proliferation in acute lymphoblastic leukemia and prevents neurodegeneration [17]. PCI34051 has been shown to inhibit HDAC8 by blocking zinc atoms in the active site of the enzyme [13]. It is probably the most widely used HDAC8-specific inhibitor in research [29]. It is known to decrease tumor growth in neuroblastoma and enhance the retinoic acid-mediated differentiation of cells [30].

Broad-spectrum HDAC inhibitors, such as Valproric acid, Trichostatin A, Vorinostat, and Apicidin, are known for their ability to induce the expression of NKG2D ligands [31]. This suggests that in lung cancer cells, the expression of NKG2D ligands is regulated at an epigenetic level via the acetylation of nuclear histones. Therefore, it is necessary to identify the types of HDAC that are involved in the expression of NKG2D ligands. Although there are several isozymes of HDACs, it was expected that only certain types of HDAC might be involved in the regulation of NKG2D ligands.

In this study, five types of selective HDAC inhibitor were used to ascertain the role of HDACs in NK cell-mediated immunity. Treatment with the inhibitor FK228 and silencing HDAC1/2 in lung cancer cells increased the expression of NKG2D ligands. However, other selective HDAC inhibitors affected the expression of NKG2D ligands only minimally. Interestingly, most of the HDAC inhibitors, with the exception of FK228, generally suppressed NKG2D ligands. We hypothesized that, apart from HDAC 1 and 2, the other HDACs would not regulate the expression of NKG2D ligands, although they might affect indirectly them through undefined secondary molecules. The small-molecule inhibitors which were used in this study could have non-specific interferences and might make unexpected alterations to the expression of NKG2D ligands.

There were some discrepancies in the expression levels of NKG2D ligands between the FK228-treated and HDAC1/2 knockdown cells. For example, although treatment with FK228 caused a significant increase in ULBP2 expression, knocking down HDAC1 or HDAC2 did not have the same effect. In contrast, the expression of ULBP1 increased significantly in A549 cells when HDAC2 was knocked down. These discrepancies might be due to the low specificity of FK228, which could inhibit HDAC1 and HDAC2 simultaneously. Although the HDAC inhibitors used in this study were known to selectively inhibit certain types of HDAC, they could also have non-specific effects on other types of HDAC. As mentioned previously, HDACs have many target proteins beside nuclear histones. The acetylation levels of these target proteins might affect a number of cellular responses, in addition to the expression of NKG2D ligands.

We hypothesized that induced NKG2D ligands might function as molecules that stimulate NK cells and promote NK cell-mediated anticancer immunity. However, we also observed a concomitant increase (approximately twofold) in the expression of HLA-ABC after treatment with FK228. These molecules could transfer negative signals to NK cells and limit the effect of activating NKG2D ligands. Here, we observed that the increase in NKG2D ligands could overcome the inhibitory signals transduced by the upregulated HLA-ABC.

In our study, inhibiting HDAC1 and HDAC2 was sufficient to induce the expression of NKG2D ligands, in a manner similar to the effect of broad-spectrum HDAC inhibitors. Inhibiting other HDACs did not induce NKG2D ligands. Therefore, HDAC1 and HDAC2 play a central role in the transcription of NKG2D ligands, and the specific inhibition of HDAC1 and HDAC2 might be a new strategy for treating lung cancer by promoting NK cell-mediated anticancer immunity.

## 4. Materials and Methods

### 4.1. Cell Lines and Reagents

This study was performed using the human non-small cell lung cancer cell lines: NCI-H23 and A549. These cell lines were obtained from the Korean Cell Line Bank (Seoul, Korea) and were maintained in an RPMI-1640 medium, which was supplemented with 10% fetal bovine serum (Gibco, Grand Island, NY, USA), 2 mM L-glutamine, 100 mg/mL streptomycin, and 100 U/mL penicillin. The NK-92 cell line was obtained from the American Type Culture Collection (Rockville, MD, USA) and maintained in an alpha-minimum essential modified medium, which was supplemented with 12.5% (*v*/*v*) fetal bovine serum, 12.5% (*v*/*v*) horse serum, 2 mM L-glutamine, 0.1 mM 2-mercaptoethanol, 200 U/mL recombinant human interleukin-2, 100 µg/mL streptomycin, and 100 U/mL penicillin. All cell lines were cultured at 37 °C in a humidified atmosphere and at 5% CO_2_. Five HDAC inhibitors were used. The inhibitor FK228 was purchased from Selleckchem (Houston, TX, USA); the inhibitors RGFP966, MC1568, and Tubacin were obtained from Sigma-Aldrich (St. Louis, MO, USA); and PCI34051 was purchased from R&D Systems (Minneapolis, MN, USA) (Figure 1).

### 4.2. Total RNA Extraction and Multiplex Reverse Transcription PCR (RT-PCR)

Total RNA extraction and qPCR were performed as described previously [6]. Briefly, total RNA was extracted from the cells using the RNeasy^®^ Mini kit (Qiagen, Hilden, Germany), according to the manufacturer’s instructions. The cDNA was synthesized from 1 µg of the extracted total RNA using 100 pmol of random primers (Takara Bio Inc., Otsu, Japan) and 100 U M-MLV reverse transcriptase (Promega Corporation, Madison, WI, USA). The cDNA was then used in the PCR reaction, which was performed using the QIAGEN Multiplex PCR kit (Qiagen, Hilden, Germany). Numerous primer pairs were used to investigate the mRNA expression levels of the following NKG2D ligands: MICA, MICB, ULBP1, ULBP2, and ULBP3. ACTB and ribosomal protein L19 (RPL19) were used as the loading control and the degradation marker, respectively (Bioneer Corporation, Daejeon, Korea). The primer sequences have been reported previously [6]. The PCR products were separated and quantified with the help of MultiNA (Shimadzu, Tokyo, Japan).

### 4.3. Flow Cytometry

The lung cancer cells were incubated with mouse anti-MICA, anti-MICB, anti-ULBP1-3, and anti-HLA-ABC monoclonal antibodies (mAbs) (R&D systems, Minneapolis, MN, USA) to determine the expression of NKG2D ligands and MHC class I molecules in lung cancer cells. Specific mAbs against NKG2D ligands and MHC class I molecules or the corresponding isotype controls were used at 1 μg in 100 μL cell suspensions. These were followed by the addition of goat anti-mouse PE-conjugated antibody (BD Phamingen Inc., San Diego, CA, USA). The analysis was performed using BD FACSCANTO II (Becton Dickinson and Company, Franklin Lakes, NJ, USA) and FlowJo software (BD Biosciences, Franklin Lakes, NJ, USA). Surface expression levels were measured as mean fluorescence intensity (MFI).

### 4.4. Flow Cytometry Analysis of the NK Cell-Mediated Cytotoxicity Assay

The NCI-H23 cells (target cells) were labeled with 5 µM CFSE for 15 min at 37 °C in a humidified incubator at 5% CO_2_ and then washed with complete medium. The NK-92 cells (effector cells) were co-cultured with CFSE-labeled NCI-H23 cells in round-bottomed 96-well plates for 4 h and at appropriate effector-to-target cell count ratios (5:1, 1:1). The DNA of dead cells was labelled with 50 µg/mL PI (Sigma-Aldrich, St. Louis, MO, USA). Dead cells were detected by means of flow cytometry. All experiments were performed in triplicate.

### 4.5. Silencing HDAC1 or HDAC2 Using siRNA Transfection

The siRNAs (siH1 and siH2, targeting HDAC1 and HDAC2, respectively) and scRNA were chemically synthesized by Life Technologies (Carlsbad, CA, USA). Lung cancer cells were transfected with 150 nM siH1 and siH2, or scRNA using the Lipofectamine RNAiMAX Reagent (Invitrogen, Carlsbad, CA, USA) according to the manufacturer’s instructions. The cells were incubated for 48 h.

### 4.6. Western Blot Analysis

Cells were lysed in a lysis buffer, and the cellular debris was removed by centrifugation at 15,000 rpm for 10 min. The proteins were separated by sodium dodecyl sulfate-polyacrylamide gel electrophoresis, transferred onto nitrocellulose membranes (Hybond-ECL, GE Healthcare, Chalfont Saint Giles, UK), blocked with 5% skimmed milk, and incubated with anti-HDAC1 and anti-HDAC2 antibodies (Abcam, 1:1000 dilution) and an anti-β actin antibody (Novus Biologicals, 1:10,000 dilution). Membranes were then incubated with horseradish peroxidase-conjugated goat anti-rabbit IgG secondary antibody (Enzo Life Science Inc., Bucharest, Romania, 1:5000 dilution) for 1 h at room temperature and illuminated by enhanced chemiluminescence (Perkin-Elmer Life Science, Waltham, MA, USA). Expression levels were measured using an Amersham Imager 680 (GE Healthcare, Chalfont Saint Giles, UK).

### 4.7. Statistical Analysis

The mean fold changes in the gene expression between groups and the standard error of the mean fold were calculated to evaluate altered gene expression. A paired Student’s t-test was performed to test for significance between groups. For all experiments, *p* < 0.05 indicates a statistically significant difference.

## Figures and Tables

**Figure 1 molecules-26-03952-f001:**
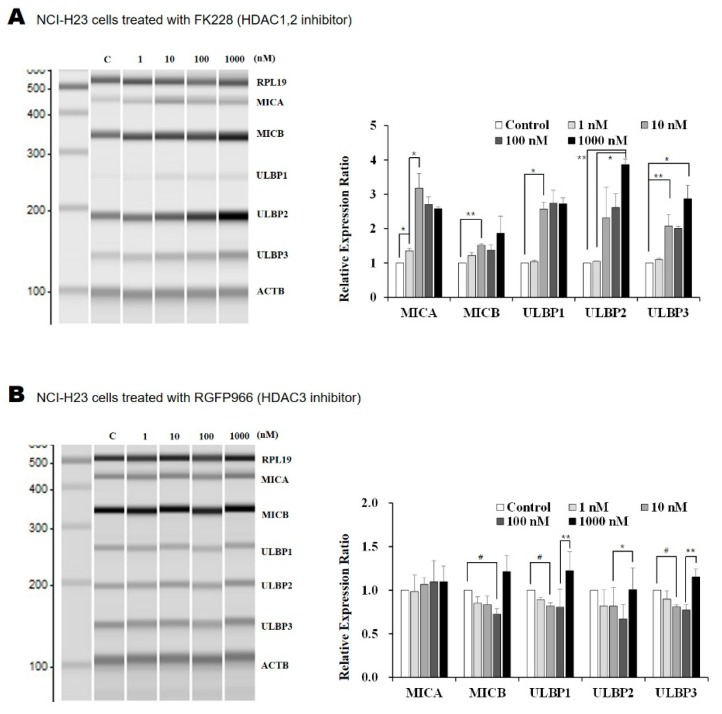
Altered transcription of NKG2D ligands after treatment with HDAC inhibitors. The transcription of NKG2D ligands was analyzed in lung cancer cells using multiplex RT-PCR. The NCI-H23 cells were each treated with 1, 10, 100, and 1000 nM of the following selective HDAC inhibitors: (**A**) FK228, (**B**) RGFP966, (**C**) MC1568, (**D**) Tubacin, and (**E**) PCI34051 for 18 h. (**F**) The A549 cells were treated with FK228 for 18 h. PCR products were separated and quantified using MultiNA. All experiments were performed in triplicate. Changes in transcription were normalized to the β-actin (ACTB) and presented as the mean fold change in comparison to the controls (treated with same amount of solvent, DMSO). * represents an increase, and # represents a decrease that has statistical significance. *p*-values less than 0.05, 0.01, and 0.001 are marked as */#, **/##, and ***, respectively.

**Figure 2 molecules-26-03952-f002:**
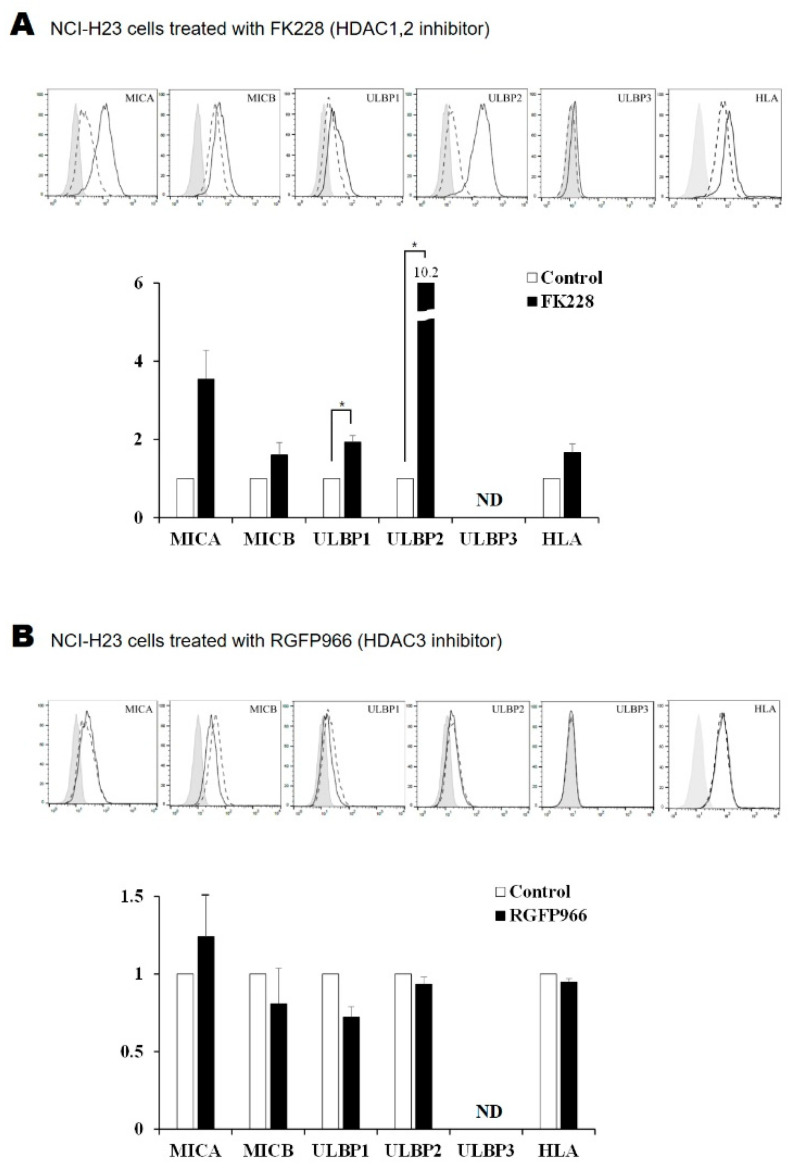
Treatment with HDAC inhibitors alters the surface expression of NKG2D ligands. Histograms show the surface expression of NKG2D ligands in NCI-H23 lung cancer cells that were treated for 24 h with 100 nM of (**A**) FK228, (**B**) RGFP966, (**C**) MC1568, (**D**) Tubacin, and (**E**) PCI34051. (**F**) The A549 cells were treated with 100 nM FK228 for 24 h. Relative expression ratios were calculated by comparing the MFI of FK228-treated cells with control cells. Solid gray, dotted, and black lines represent the isotype control, media-treated control, and HDAC inhibitor-treated samples, respectively. * represents a statistically significant increase and # represents a statistically significant decrease. *p*-values less than 0.05, 0.01, and 0.001 are marked as *, **/##, and ### symbols, respectively. ND means not detected.

**Figure 3 molecules-26-03952-f003:**
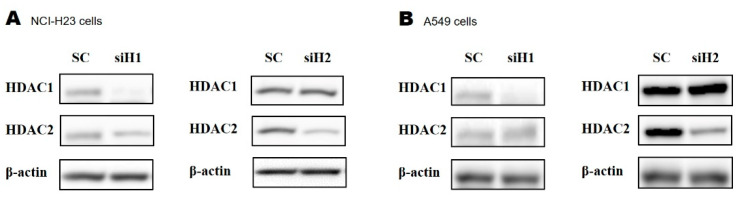
Silencing HDAC1 or HDAC2 using specific siRNA. The (**A**) NCI-H23 and (**B**) A549 cells were incubated for 48 h after transfection with 150 nM siH1 (siRNA targeting HDAC1), siH2 (siRNA targeting HDAC2), and scrambled RNA (scRNA). The levels of HDAC1 and HDAC2 expression were analyzed by Western blotting using anti-HDAC1 and anti-HDAC2 monoclonal antibodies. β-actin was used as the loading control.

**Figure 4 molecules-26-03952-f004:**
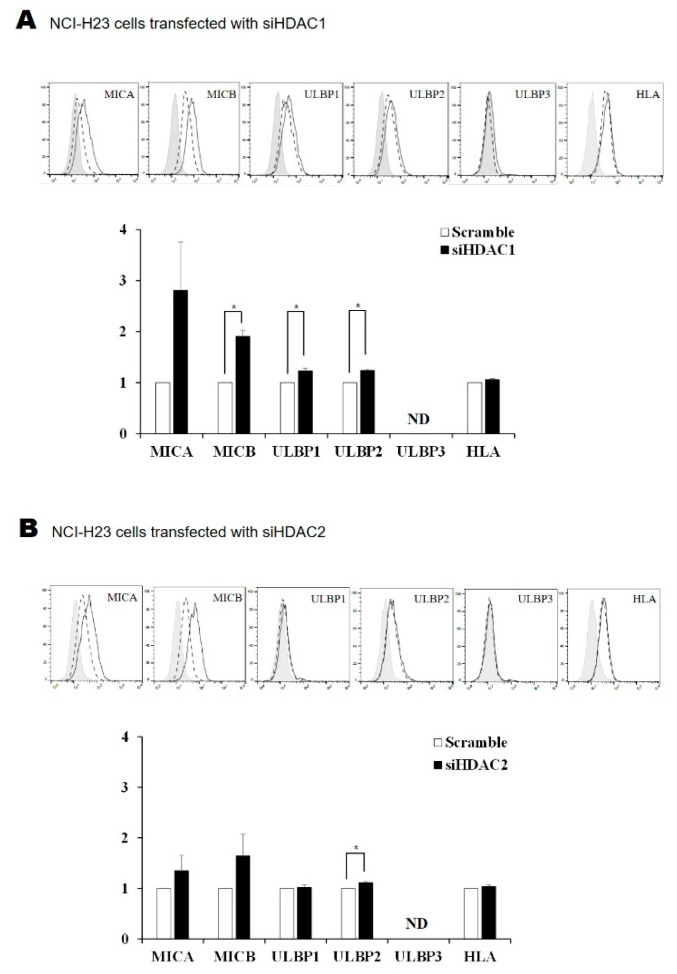
Surface expression of NKG2D ligands after HDAC1 or HDAC2 silencing. Surface protein expression of NKG2D ligands was analyzed using flow cytometry 48 h after transfecting NCI-H23 (**A**,**B**) and A549 cells (**C**,**D**) with siH1 or siH2. The solid gray, dotted, and black lines represent isotype control, scRNA, and siRNA, respectively. * represents a statistically significant increase. *p*-values less than 0.05 and 0.01 are marked as * and ** symbols, respectively. ND means not detected.

**Figure 5 molecules-26-03952-f005:**
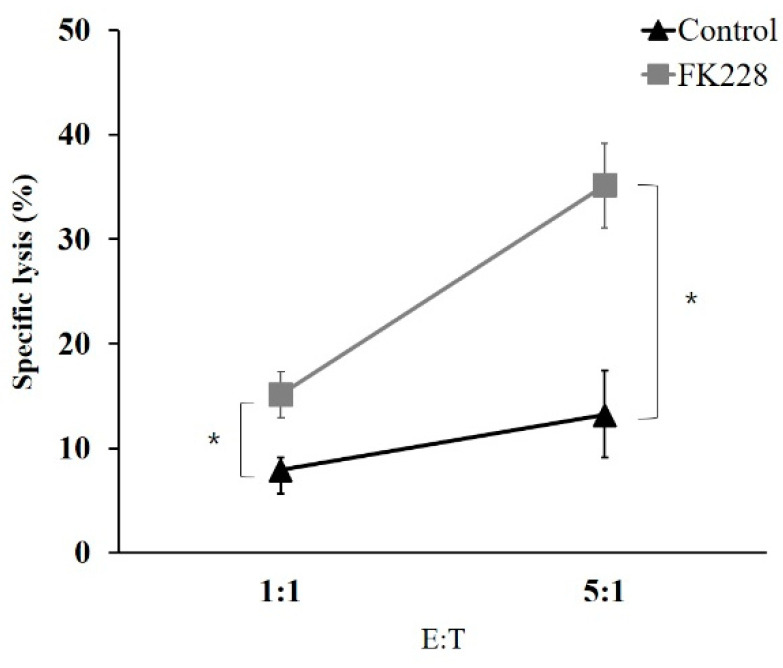
Selective cell death by NK cell cytotoxicity after treatment with FK228. NCI-H23 cells were treated with FK228 for 24 h at 100 nM and then stained with carboxyfluorescein succinimidyl ester (CFSE). The target cells were co-cultured with NK-92 cells for 4 h at effector-to-target ratios of 1:1 and 5:1. Flow cytometry was used to measure the proportion of PI-positive cells, which indicated NK cell-mediated lysis. * represents a statistically significant increase. *p*-values less than 0.05 is marked as * symbols.

**Table 1 molecules-26-03952-t001:** The 18 types of HDAC and 6 selective HDAC inhibitors.

The Classification of HDACs and Their Selective Inhibitors				
Zinc-dependent	Class I	HDAC1	FK228	Resveratrol
		HDAC2		
		HDAC3	RGFP966	
		HDAC8	RGFP966	
	Class IIa	HDAC4	MC1568	
		HDAC5		
		HDAC7		
		HDAC9		
	Class IIb	HDAC6	Tubacin	
		HDAC10		
	Class IV	HDAC11		
DAD-dependent	Class III	Sirtuin		

## Data Availability

The data and all samples could be available and provded by a corresponding author (Bae, J) under permission of authors.

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
