# Peer review of "Differential Effects of Histone Deacetylases on the Expression of NKG2D Ligands and NK Cell-Mediated Anticancer Immunity in Lung Cancer Cells"

_molecules, 2021, doi:10.3390/molecules26133952_

Round 1
Reviewer 1 Report
The topic was well chosen and explained and the use of multiple HDAC inhibitors and comparing in between their effects enriched the content and the credibility of the results
corrections:
- Previous studies suggest (line 178)
Author Response
Thanks for your detailed review.
We have tried to do as the other reviewer's suggestion and some descriptions have been corrected in the text.
Reviewer 2 Report
Comments:
NK cells are involved in the immune surveillance of cancer and persistent NK cell activity is associated with a good prognosis of patients with lung cancer. NK cells express the major activating receptor NKG2D, which transduces activating signals to immune cells upon binding to the NKG2D ligands on lung cancer cells. In this manuscript, Cho et al found that specifically inhibiting HDAC1 and HDAC2 could induce the expression of NKG2D ligands in lung cancer cells and improve the NK cell-mediated anti-cancer immunity. However, some critical experiments are required to validate the NK cell-mediated anti-cancer immunity. For example, experiments to show the cytokines secreted by NK cells, NK cell conjugation ability with lung cancer cells, or mice model should be performed.
Minor points:
- There are 18 potential human HDACs grouped into four classes, including Class I (HDAC1, HDAC2, HDAC3, HDAC8), Class IIa (HDAC4, HDAC5, HDAC7, HDAC9), Class IIb (HDAC6, HDAC10), Class III (sirtuin1-7) and Class IV (HDAC11).
- The word tubulin in line 83 should be changed to Tubacin.
- The symbol should be changed to #.
Author Response
Thanks for your detailed review. We have tried to do as your suggestion and some descriptions have been corrected in the text. We'll send you a detailed response to your suggestion below.
Point: Some critical experiments are required to validate the NK cell-mediated anti-cancer immunity. For example, experiments to show the cytokines secreted by NK cells, NK cell conjugation ability with lung cancer cells, or mice model should be performed.
Response:
Sure, measurement of killing cytokines, such as granzyme and perforin, derived from NK cells was helpful to support our study. However, cytotoxicity of NK cells was mediated by not only the killing cytokines but also several surface ligands which engaging apoptosis of cancer cells, such as FasL, TRAIL and TNF. We wish to measure the activity of NK cells reagrdless mechanisms. We have plan to perform further study to investigate the role of cytokines and death ligands.
Minor points:
- There are 18 potential human HDACs grouped into four classes, including Class I (HDAC1, HDAC2, HDAC3, HDAC8), Class IIa (HDAC4, HDAC5, HDAC7, HDAC9), Class IIb (HDAC6, HDAC10), Class III (sirtuin1-7) and Class IV (HDAC11).
Response 1: Until now, it was identified 7 isofroms of sirtuin (sirt1-7) which have deacetylase activity (Ramu Manjula et al, 2021, Frontiers in Phamacology). We accepted your point and corrected the test in the manuscript. - The word tubulin in line 83 should be changed to Tubacin. Response 2: we mistook and corrected the text in the manuscript.
- The symbol should be changed to #. Response 3: We wish to distingish upregulation and downregulation. So, we represent the marks * as an increase, # as a decrease that has statistical significance.
Reviewer 3 Report
Cho et al have described their study regarding the link between HDAC function and NKG2D ligand expression in lung cancer cell lines. A selective HDAC1/2 inhibitor, as well as silencing HDAC1/2 using specific siRNA resulted in increased NKG2D ligand expression. They have also demonstrated increased susceptibility of cell lines to NK cells.
Influencing anti-tumor immunity is in the limelight of oncological treatment. The manuscript is well written, easy to understand. The methods used appear sound, the conclusions are backed up by results. References are adequate and up-to-date.
Few comments are the following:
The Figure legends are difficult to separate from the main text, however, this may be a feature of the review format.
Figure legend for Figure 5 should include effector-to-target ratio as an explanation of E: T.
The authors state that increase of MICA and ULBP1 expression following FK228 treatment was dose-dependent (Page 3, lines 85-87). Why was ULBP2 increase not regarded as similarly dose-dependent?
Seeing no significant fold change of NKG2D ligands after various treatments in some cell lines does not necessary indicate there is no effect of the treatment; NK2GD ligand expression may have been already high. The authors should at least comment on this issue in the discussion chapter. Were the expression of NK2GD ligands similar in the two investigated cell lines without any treatment? Why was only FK228 tested on A549 cells? It is interesting to see no expression of ULBP3 was detected at the protein level in NCI-H23 cells, while transcripts were detected and have shown significant changes upon various treatments. Two previous papers that was coauthored by one of the coauthors of the present paper (Med Sci Monit. 2020;26:e926395-1–e926395-10 PMID: 33139690 and Exp Mol Med . 2006 Oct 31;38(5):474-84. PMID: 17079863) did report ULBP3 expression with flow cytometry on NCI-H23 cells. What was the reason for not detecting ULBP3 expression in the present study?
Author Response
Thanks for your detailed review. We have tried to do as you suggested and some descriptions have been corrected in the text. We'll send you a detailed response to your suggestion below.
Point 1: The Figure legends are difficult to separate from the main text, however, this may be a feature of the review format.
Response 1: This manuscript was made for writing. We expect our editors to edit them in proper layout.
Point 2: Figure legend for Figure 5 should include effector-to-target ratio as an explanation of E: T.
Response 2: Sure, we inserted a comment the E/T ratio in figure 5 legend.
Point 3: The authors state that increase of MICA and ULBP1 expression following FK228 treatment was dose-dependent (Page 3, lines 85-87). Why was ULBP2 increase not regarded as similarly dose-dependent?
Response 3: Sure, the alteration of ULBP2 of NCI-H23 cells also showed the dose-dependent manner. However, we wish to describe the results conservatively. The expression of ULBP2 between control and 1000 nM treated group is upregulated significantly. The expression of ULBP2 between 1 nM and 1000 nM treated group is upregulated significantly also. However, that among treated groups was not changed with statistically significant.
Point 4: Seeing no significant fold change of NKG2D ligands after various treatments in some cell lines does not necessary indicate there is no effect of the treatment; NK2GD ligand expression may have been already high. The authors should at least comment on this issue in the discussion chapter.
Response 4: Sure, most of HDAC inhibitors in this study showed no significant changes in the expression of NKG2D ligands. We guessed that these HDACs could affect indirectly them through secondary molecules. Furthermore, although the inhibitors have a selective effect to specific kinds of HDACs, they could affect undefined molecules due to non-specific inhibition. We described this issue in discussion section additionally.
Point 5: Were the expression of NK2GD ligands similar in the two investigated cell lines without any treatment? Why was only FK228 tested on A549 cells?
Response 5: We already knew that the inhibition of Pan-HDAC could induce NKG2D ligands. Most of HDAC inhibitors except FK228 showed the suppressive effect on the expression of NKG2D ligands in NCI-H23 cells. It was not necessary to confirm the negative effects of other HDAC inhibitors in A549 cells.
Point 6: It is interesting to see no expression of ULBP3 was detected at the protein level in NCI-H23 cells, while transcripts were detected and have shown significant changes upon various treatments. Two previous papers that was coauthored by one of the coauthors of the present paper (Med Sci Monit. 2020;26:e926395-1–e926395-10 PMID: 33139690 and Exp Mol Med . 2006 Oct 31;38(5):474-84. PMID: 17079863) did report ULBP3 expression with flow cytometry on NCI-H23 cells. What was the reason for not detecting ULBP3 expression in the present study?
Response 6: That was the part that perplexed us the most in this study. The expression of ULBP3 did not show any significant difference from the isotype control even after several experiments. So, very tiny alterations make huge fold changes in ULBP3. We decided to treat the ULBP3 as an undetected molecule. There was no problem in the detection of ULBP3 in cells other than NCI-H23 (eg, A549). Therefore, it is not considered to be a problem with the antibody or experimental technique. One possible assumption is that the characteristics of genes responding to stress may be affected by minute differences in cell culture conditions and medium composition (difference in FBS lot). In some of the previous experiments pointed out by the reviewer, the expression level of ULBP3 was slightly different depending on the experimenter.
Round 2
Reviewer 2 Report
The authors rvised the manuscript, although no further results were provided.